# Systemic Inflammatory Predictors of In-Hospital Mortality in COVID-19 Patients: A Retrospective Study

**DOI:** 10.3390/diagnostics12040859

**Published:** 2022-03-30

**Authors:** Bartosz Kudlinski, Dominika Zgoła, Marta Stolińska, Magdalena Murkos, Jagoda Kania, Pawel Nowak, Anna Noga, Magdalena Wojciech, Gabriel Zaborniak, Agnieszka Zembron-Lacny

**Affiliations:** 1Department of Anaesthesiology, Intensive Care and Emergency Medicine, Collegium Medicum University of Zielona Góra, 65-417 Zielona Gora, Poland; dominika.zgola@gmail.com (D.Z.); mrtstolinska@gmail.com (M.S.); magdalena.murkos@gmail.com (M.M.); kania.jag@gmail.com (J.K.); 2Student Research Group, University of Zielona Gora, 65-417 Zielona Gora, Poland; nowak.pawel970@gmail.com (P.N.); annmaria1605@gmail.com (A.N.); 3Department of Mathematical Statistics and Econometrics, Faculty of Mathematics, Computer Science and Econometrics, University of Zielona Góra, 65-516 Zielona Gora, Poland; m.wojciech@wmie.uz.zgora.pl (M.W.); gabrielzaborniak@gmail.com (G.Z.); 4Department of Applied and Clinical Physiology, Collegium Medicum University of Zielona Gora, 28 Zyty Str., 65-417 Zielona Gora, Poland; a.zembron-lacny@cm.uz.zgora.pl

**Keywords:** age, comorbidities, lymphocytes, neutrophils, survival analysis

## Abstract

The purpose of this study was to investigate whether routine blood tests and clinical characteristics can predict in-hospital mortality in COVID-19. Clinical data of 285 patients aged 59.7 ± 10.3 yrs. (males *n* = 189, females *n* = 96) were retrospectively collected from December 2020 to June 2021. Routine blood tests were recorded within the 1st hour of admission to hospital. The inflammatory variables, such as C-reactive protein (CRP), procalcitonin (PCT), neutrophils–lymphocyte ratio (NLR) and the systemic inflammatory index (SII), exceeded the reference values in all patients and were significantly higher in deceased patients (*n* = 108) compared to survivors (*n* = 177). The log-rank test for comparing two survival curves showed that patients aged ≥60.5 years, with PCT ≥ 0.188 ng/mL or NLR ≥ 11.57 10^3^/µL were at a greater risk of death. NLR demonstrated a high impact on the COVID-19 mortality (HR 1.317; 95%CI 1.004–1.728; *p* < 0.05), whereas CRP and SII showed no effect (HR 1.000; 95%CI 1.000–1.004; *p* = 0.085 and HR 1.078; 95%CI 0.865–1.344; *p* = 0.503, respectively). In the first Polish study including COVID-19 patients, we demonstrated that age in relation to simple parameters derived from complete blood cell count has prognostic implications in the course of COVID-19 and can identify the patients at a higher risk of in-hospital mortality.

## 1. Introduction

From March 2020, all countries were affected by coronavirus disease 2019 (COVID-19), with over 420 million confirmed cases leading to 5.9 million deaths [1]. In Poland, COVID-19 has been confirmed in over 5.6 million cases and has resulted in 110,517 deaths [2]. COVID-19 is an infectious disease with a wide range of clinical symptoms, from asymptomatic to mildly symptomatic and severe forms, pointing to a major role of the host response to SARS-CoV-2 (severe acute respiratory syndrome coronavirus 2) [3]. Patients may show the following symptoms: fever, high temperature, cough, myalgia, sputum production, headache, haemoptysis, diarrhoea, dyspnoea, and, in some cases, acute respiratory distress syndrome (ARDS), acute cardiac injury, or secondary infection [4]. Most of the infections are not severe, but 81% are mild, 14% of the cases are severe (with dyspnoea, hypoxia, or >50% lung involvement on diagnostic imaging), and 5% develop a critical disease with respiratory failure, shock, or multiorgan dysfunction [5]. The risk of death from COVID-19 strongly depends on the age and previous health status. Older patients are much more prone to critical and fatal disease outcomes, especially with comorbidities, such as cardiovascular diseases, hypertension, chronic kidney disease, diabetes, and pulmonary disease [6,7]. These associations may contribute to the course of COVID-19 or determine the time of a patient’s death. According to Elezkurtaj et al. [7], the immediate causes of death were directly linked to the lung damage initiated by SARS-CoV-2 infection and, in most cases, were unrelated to pre-existing health conditions and comorbidities. In addition, the authors also supported the idea that patients who died of COVID-19 appear to have lost considerable lifetime, independent of their age [7].

Until now, little is known about the relationship between clinical patterns, systemic non-specific markers of inflammation, and the immune response. Previously reported modifications in severe forms of COVID-19 showed increased levels of C-reactive protein (CRP), procalcitonin (PCT), ferritin, lactate dehydrogenase (LDH), interleukin 6 (IL-6), D-dimers, cardiac troponin, and renal markers, whereas decreased levels of albumin turned out to predict mortality in hospitalised COVID-19 patients [8,9,10,11]. Severe cases of COVID-19 demonstrated lower lymphocyte numbers, higher numbers of leukocytes, and a greater neutrophil–lymphocyte ratio as well as smaller proportions of monocytes and eosinophils [8]. In the past few years, many studies have found that the combinations of the haematological components of the systemic inflammatory response, such as the neutrophil–lymphocyte ratio (NLR), the platelet–lymphocyte ratio (PLR), the lymphocyte–monocyte ratio (LMR), and the systemic immune inflammation index (SII) were effective prognostic indicators in patients with a variety of cancers [12,13,14,15], obesity [16], coronary artery disease [17,18], diabetes [19,20], acute ischaemic stroke [21,22], and also COVID-19 [23,24,25,26,27]. The components of these easily calculated parameters are readily available, inexpensive, and routinely measured in daily practice as part of the complete blood count report [28]. The calculation of these haematological components of the systemic inflammatory response may provide clinicians with a further valuable tool for clinical risk stratification. Therefore, our aim was to investigate and compare the prognostic impacts of CRP, PCT NLR, PLR, LMR, and SII biomarkers in laboratory-confirmed COVID-19 cases as well as to explore the most useful diagnostic biomarkers and optimal cut-off values in COVID-19 patients to predict in-hospital mortality.

## 2. Materials and Methods

### 2.1. Study Population

This retrospective study analysed a database including 285 patients over 18 years of age that were consecutively admitted to the intensive care unit (ICU) at the University Hospital in Zielona Gora (Poland) from December 2020 to June 2021 (Table 1). The patients were classified as severe acute respiratory distress syndrome (ARDS). The ratio of partial pressure arterial oxygen (PaO_2_) to fraction of inspired oxygen (FIO_2_) was 121.45 ± 82.15 mm Hg for survivors and 104.28 ± 70.33 mm Hg for non-survivors (*p* > 0.05). The study protocol was approved by the Bioethics Commission at the Regional Medical Chamber of Zielona Gora, Poland (No. 21/157/2021), in accordance with the Declaration of Helsinki.

COVID-19 patients were staged according to the clinical risk score of Liang et al. [29]. The general admission criteria included characteristic radiographic results and/or shortness of breath, which was defined as tachypnoea and/or low oxygen pulse in the absence of an alternative diagnosis. The criteria for ICU admission included low oxygen pulse despite supplementary oxygen with a non-rebreather mask, sepsis per the sequential organ failure assessment (SOFA) criteria and/or required mechanical ventilation, a vasopressor-requiring shock, and unexplained confusion. The diagnosis of SARS-CoV-2 infection was confirmed by reverse-transcriptase polymerase chain reaction (RT-PCR) of a nasopharyngeal swab. All patients received standard supportive care, including low-molecular-weight heparin, statins, and supplementary oxygen on demand. Patients with severe COVID-19, with oxygen saturation of ≤90%, additionally received corticosteroids, while critically ill patients received tocilizumab.

### 2.2. Blood Collection and Haematological and Biochemical Variables

Blood samples were collected within one hour of admission to hospital using S-Monovette-EDTA K2 tubes (Sarsted AG & Co. KG, Nümbrecht, Germany). Haematological parameters, including the total white blood cell count (WBC), red blood cell count (RBC), platelet count (PLT), differential white cell count (neutrophils, lymphocytes, monocytes, eosinophils, and basophils), and haemoglobin concentration (HB) were determined with a Sysmex XN-1000 analyser (Sysmex Europe Gmbh, Norderstedt, Germany). The neutrophil–lymphocyte ratio (NLR), platelet–lymphocyte ratio (PLR), lymphocyte–monocyte ratio (LMR), and systemic immune inflammation index (SII) were calculated and compared to the reference values according to Luo et al. [28]. In brief, the formulas are as follows: NLR = neutrophils (10^3^/µL)/lymphocytes (10^3^/µL); PLR = platelets (10^3^/µL)/lymphocytes (10^3^/µL); LMR = lymphocytes (10^3^/µL)/monocytes (10^3^/µL); systemic inflammatory index SII = (platelets (10^3^/µL) × (neutrophils (10^3^/µL)/lymphocytes (10^3^/µL)). The levels of C-reactive protein (CRP), procalcitonin (PCT), creatine, vitamin D, and D-dimers were determined using Cobas C501 and Cobas e601 analysers (Roche Basel, Switzerland).

### 2.3. Statistical Analysis

Statistical analyses were performed using RStudio, version 4.1.2 [30]. The variables were reported as mean values ± standard deviations (SD) or medians with interquartile ranges (iqr). The statistical significance of intergroup differences was compared through a Kruskal–Wallis rank sum test for continuous variables and χ^2^ test, or Fisher’s exact test was used to compare categorical variables. The predictive value of variables was evaluated by measuring the area under the receiver operating characteristic curve (ROC curve). The optimal threshold value for clinical stratification (cut-off value) was obtained by calculating the Youden index. Survival curves were plotted by using the Kaplan–Meier method and were compared using the log-rank test. A Cox proportional hazards regression (HR) was performed for both the univariate and multivariate analyses. Additionally, Spearman’s rank correlation (r_s_, Spearman rank correlation coefficient) was used to assess the relationships between inflammatory markers. Statistical significance was set at *p* < 0.05.

## 3. Results

### 3.1. Study Population

A total of 285 patients were included in this study, with 66.3% being males aged 58.5 ± 10.6 years and 33.7% being females aged 60.9 ± 10.0 years. The survivors constituted 62% of all patients, and they were significantly younger (57.0 ± 10.7 years) compared to the deceased patients (63.0 ± 9 years). The risk of death from COVID-19 for patients aged ≥60.5 years was found to be greater than for the younger ones. Moreover, the probability of survival decreased considerably faster in older patients than young patients during hospitalisation (Figure 1a). The number of hospitalised men was two-fold higher than women, and men also predominated in the deceased patients. The number of patients with comorbidities amounted for 201, and cardiovascular diseases were predominant comorbidities (>70%). The incidence of concomitant diseases in survivors and non-survivors was similar (Table 1).

### 3.2. Haematological and Biochemical Variables

The haematological markers were found to fall within the referential ranges, and they did not differ significantly between groups, whereby MCH and MCHC were significantly lower in survivors when compared to the deceased patients. Similarly, CRP, PCT, creatine, and D-dimer levels exceeded the reference values in both groups, and they were significantly higher in the deceased patients. In the case of D-dimers, the values were elevated approx. 3-fold in deceased patients and were very diverse in both groups. Vitamin D levels were below the reference values, but they did not differ between groups. This suggests that a low intake of Vitamin D is not related to the severity of COVID-19 (Table 2). The neutrophil count exceeded the reference values in all patients, and a 2-fold increase was recorded in the deceased patients. The counts of lymphocytes, monocytes, and platelets did not differ between the patient groups. The NRL and SII exceeded reference values in both groups, and they were significantly elevated in the deceased patients (Table 3). The NRL and SII were highly interrelated (r_s_ = 0.872, *p* < 0.001), and the NRL significantly correlated with the age and other markers of inflammation in the survivors (Table 4).

The results of the ROC analysis of the age and inflammation markers ranged between 0.6 and 0.7, indicating a potential diagnostic value for clinical prognosis. Furthermore, the sensitivity, specificity, and positive and negative predictive values of these inflammation markers were calculated to obtain the optimal threshold values, which corresponded to 60.5 years for age, 140.20 mg/L for CRP, 0.188 ng/mL for PCT, 11.57 10^3^/µL for the NLR, and 2058 10^3^/µL for the SII (Table 5). Moreover, the Kaplan–Meyer survival curves showed that elder patients (≥60.5 years) and those with PCT or NLR higher than the optimal threshold value had a significantly higher probability of death (Figure 1a,c,d). With regard to age, the Kaplan–Meier survival curves were significantly different (log-rank *p* < 0.001), and the median survival was 32 days for patients aged <60.5 years and 17 days for patients ≥60.5 years (Figure 1a).

The Cox model confirmed that age, PCT, and NLR levels above the optimal threshold values significantly increased the risk of death (Table 6). Among the analysed variables, the NLR demonstrated the highest impact on the mortality rate (HR > 2, *p* < 0.01), whereas CRP was found to have no effect (HR 1.000, *p* > 0.05).

## 4. Discussion

A rapid clinical diagnosis is crucial in symptomatic treatment, urgent access to the ICU, and patient isolation to prevent the transmission of COVID-19. Despite some widely recognised challenges, such as long testing timed and no PCR laboratory equipment in some hospitals, the PCR test is still the gold standard for COVID-19 analysis [31]. Other widely used techniques, such as biochemical and complete blood count analysis, might be faster, easy-to-measure and low-cost techniques that facilitate the diagnosis and prognosis of COVID-19 [26]. However, these common techniques do not allow an accurate the diagnosis, while the integration of inflammatory status measurements with overall survival models may provide physicians with a valuable tool for clinical risk stratification in COVID-19.

Age is one of the most frequently reported factors associated with a severe form or fatality of COVID-19 [32]. A potential explanation is the association observed between an older age and a decline in the immune competence associated with the increased susceptibility to a number of chronic diseases as well as an impaired response to vaccination. The immunosenescence-related disproportion in CD4+ and CD8+ T lymphocytes increases the risk of infectious diseases and contributes to cardiovascular, metabolic, autoimmune, and neurodegenerative diseases [33]. The present study confirms that the risk of death from COVID-19 in patients aged ≥60.5 years is greater than in younger ones. Moreover, the probability of survival decreases considerably faster in older patients than young patients during the hospitalisation period (Figure 1a). The experience gained during the Italian epidemic pointed to the patients’ age as one of the most important risk factors for COVID-19 mortality [34], and this conclusion was supported by findings from other reports [32,35]. However, a recent study demonstrated that patients who died of COVID-19 appear to have lost considerable lifetime, independent of their age [7].

In comparison to the data reported by Fumagalli and al. [11], younger age, low paO_2_/FiO_2_ values (121.45 ± 82.15 mm Hg for survivors and 104.28 ± 70.33 mm Hg for non-survivors (*p* > 0.05)) and the necessary application of mechanical ventilation in our study patients seem to have resulted from a delayed and/or limited access to medical assistance.

Males dominated in our intubated patients and constituted 66.3%, which is consistent with other reports. The studies from China, South Korea, and Italy and autopsy findings from Germany have reported that males accounted for 59–75% of COVID-19 patients [36,37,38,39,40,41]. A large analysis of COVID-19 adults hospitalised at US academic centres showed a higher rate of respiratory support by intubation, longer hospitalisation, and a higher death rate in males when compared to females [42]. In our study, men constituted 66.3% of all patients and 69.4% of the deceased patients, which confirms that the male sex is a predictor of higher morbidity and mortality from COVID-19 [40,42,43,44]. Sex differences in both the innate and adaptive immune systems have been reported and may account for the female advantage in COVID-19. Within the adaptive immune system, higher numbers of CD4+ T cells, more robust CD8+ T cell cytotoxic activity [45], and increased B cell production of immunoglobulin were identified in females when compared to males [44]. Moreover, women demonstrated more severe local and systemic side effects and produced higher antibody titres in response to the seasonal influenza vaccination [46]. These data have implications for the clinical management of COVID-19 and highlight the importance of sex as a variable to be considered in fundamental and clinical studies [44].

It has become clear that cardiovascular comorbidities are a significant risk factor of COVID-19 hospitalisation and mortality [40]. In our study, cardiovascular diseases, such as arterial hypertension, were the most common comorbid conditions (Table 1). Recently, we found that hypertension has the greatest impact on the T cell subpopulation and that it determines the direction of changes in CD4+/CD8+ ratio in the Polish population over 60 years of age [45]. A retrospective observational study by Gao et al. [47] showed a two-fold increase in COVID-19 mortality in hypertensive individuals, highlighting a clear relationship between hypertension and COVID-19. On the other hand, Elezkurtaj et al. [7] argued that the immediate causes of death were directly linked to the lung damage initiated by SARS-CoV-2 infection, and, in most cases, they were unrelated to pre-existing health conditions and comorbidities. Although further studies are needed, these data already indicate that cardiovascular diseases contribute to the severity of COVID-19. Likewise, the inflammation resulting from SARS-CoV-2 infection could provide an explanation for the increased incidence of cardiovascular events [40].

Circulating CRP and other inflammation mediators have been implicated in cardiovascular diseases and have also been investigated as independent predictors of prognosis in COVID-19 [9,11,48]. CRP is a non-specific acute-phase protein induced by IL-6 in the liver and a marker of inflammation, bacterial or viral infection, and tissue damage. However, CRP is not only just a marker of infection, but it also plays an active role in the inflammatory process. Key areas of the inflammatory response to CRP-mediated infections include the complement pathway, apoptosis, phagocytosis, NO release, and pro-inflammatory cytokine production, particularly IL-6 and TNFα [49]. Recently, more importance has been attached to CRP as one of the very first markers to indicate COVID-19 and its severity, despite CRP being non-specific in nature [8]. In our study, the CRP level was found to exceed the reference values in all patients, and it was significantly higher in the deceased patients. However, the Kaplan–Meier survival curves and Cox proportional hazard model analysis showed that CRP could not be used as an independent factor to predict the severity of COVID-19 (Figure 1b, Table 6).

PCT is a precursor of the calcitonin hormone that contributes to the maintenance of calcium homeostasis. Systemic PCT production by thyroid parafollicular C cells is caused by bacteriocins and by pro-inflammatory cytokines, such as TNFα, IL-1β, and IL-6. PCT synthesis is inhibited by high IFNγ levels due to viral infection [50]. Most studies reported that secondary bacterial infections in COVID-19 patients were associated with increased mortality, and they disproportionately affected critically ill patients [51,52,53]. Wan et al. [54] reported that despite significantly elevated PCT, no evidence of bacterial infection in severe COVID-19 patients was identified. Actual bacterial infection rates were recorded at 7–14% [53]. PCT rise was indicated as a response to systemic inflammatory dysregulation or hyperinflammation, most notably the suppression of IFNγ, rather than as a result of bacterial pathogens [55,56]. According to the COVID-19 guidelines prepared by National Institute for Health and Care Excellence [57], there is insufficient evidence to recommend routine PCT testing to guide decisions on antibiotic therapy. On the other hand, the implementation of PCT-based decision on antibiotic therapy can decrease the consumption of antibiotics in various COVID-19 populations [58]. Brechot et al. [59] showed that antibiotics could be discontinued when PCT was <1 ng/mL or decreased by >35% from the baseline value and clinical signs of infection resolved after at least 3 days of therapy. Williams et al. [60] reported reduced antibiotics consumption in patients with PCT ≤0.25 ng/mL with no increase in mortality. We observed that the PCT level exceeded the reference values in 36% of our patients. However, a significantly higher probability of mortality from COVID-19 was identified in the patients with a PCT cut-off value ≥0.188 ng/mL. The Cox analysis confirmed that PCT ≥0.188 ng/mL was a significantly worse prognostic factor, with a hazard ratio of 1.296 (*p* < 0.01). A similar cut-off value ≥0.1 ng/mL for PCT appeared to be associated with poor overall survival in several types of cancer [61,62,63].

The pathophysiology of severe COVID-19 infection is marked by elevated numbers of neutrophils in the nasopharyngeal epithelium, distal parts of the lungs, and also in blood [64,65,66]. Additionally, COVID-19-related disproportion in CD4+ and CD8+ T lymphocytes led to increases in the NLR value, which was reported to be a more sensitive biomarker of inflammation than the individual levels of neutrophils and lymphocytes [23]. In our study, lymphopenia was observed in 28% of patients, whereas neutrophilia was observed in 62% of patients. The determination of the neutrophil–lymphocyte ratio made it possible to compare our patients. The NLR levels exceeded the reference values in both groups and were significantly higher in the deceased patients. Furthermore, a multivariate Cox model analysis showed a significantly higher probability of death, with s hazard ratio of 2.128 (*p* < 0.001) in patients with NLR ≥11.57 10^3^/µL. Previous studies showed that elevated NLR levels could be considered as independent biomarkers of poor clinical outcomes and could be associated with the severity of COVID-19 [23,24,25,26,27]. Actually, NLR and lymphocyte subset determinations are helpful in the early screening of critical cases, diagnosis, and treatment of COVID-19 [5]. 

Numerous previous studies, including the paper by Giannini and al. [67], have already discussed the significance of the D-dimer level as an independent predictor of mortality in severe cases of ARDS in the course of SARS-CoV-2. The outcomes obtained in our study (Table 2) are consistent with these reports. By contrast, however, the Vitamin D level was not found to be as significant in our study population.

## 5. Conclusions

The present study demonstrated that some simple parameters derived from complete blood cell analysis have prognostic implications in the course of COVID-19, enable the identification of patients at a higher risk of in-hospital mortality, and confirm the preliminary observations on the suitability of the neutrophil–lymphocyte ratio for prognostic stratification.

## 6. Limitations

The limitations of the study include its single-centre and retrospective design. Furthermore, some specific variables that could impact the survival probability were not included in the log-rank test, which is another shortcoming of the study.

## Figures and Tables

**Figure 1 diagnostics-12-00859-f001:**
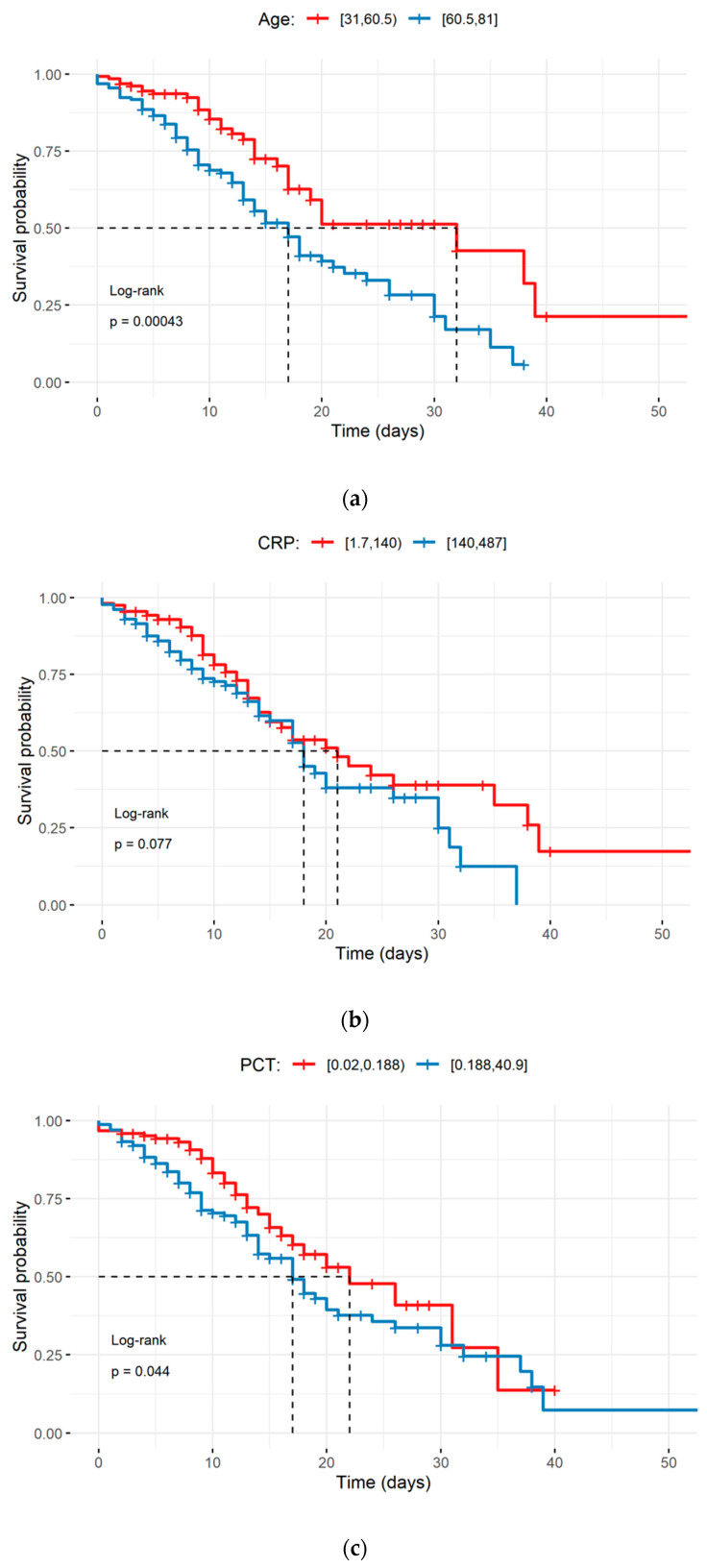
Kaplan-Meier survival curves during hospitalisation of COVID-19 patients with different cut-off values of the age and systemic inflammation markers: (**a**) age; (**b**) C-reactive protein; (**c**) procalcitonin; (**d**) the neutrophil–lymphocyte ratio; (**e**) the systemic immune inflammation index. The dotted line designates median survival. The survival comparison was performed using the log-rank test.

**Table 1 diagnostics-12-00859-t001:** The demographic and clinical data.

Characteristics	Total	Survivors	Non-Survivors	*p*-Value
Number of subjects	285	177	108	
Median age (years)	62.0	59.0	65.0	
Mean age (years)	59.3	57.0	63.0	<0.001
Number of males, n (%)	189 (66.3)	114 (64.4)	75 (69.4)	<0.001
Number of females, n (%)	96 (33.7)	63 (35.6)	33 (30.6)	<0.001
Obesity, n (%)	134 (47.7)	88 (50.0)	46 (43.8)	0.315
Cigarette smokers, n (%)	20 (7.0)	9 (5.1)	11 (10.3)	0.097
Comorbidities, n (%):				
Arterial hypertension	153 (55.2)	92 (52.6)	61 (59.8)	0.243
Coronary heart disease	26 (9.4)	11 (6.3)	15 (14.7)	0.020
Heart failure	9 (3.2)	5 (2.9)	4 (3.9)	0.729
Arterial fibrillation	10 (3.6)	4 (2.3)	6 (5.9)	0.180
Chronic kidney disease	8 (2.9)	3 (1.7)	5 (4.9)	0.149
Chronic obstructive pulmonary disease	9 (3.2)	3 (1.7)	6 (5.9)	0.080
Asthma	18 (6.5)	10 (5.7)	8 (7.8)	0.488
Diabetes mellitus	57 (20.7)	30 (17.1)	27 (26.7)	0.058
Thyroid disease	24 (8.7)	17 (9.7)	7 (6.9)	0.416
Immunosuppression	2 (0.7)	0 (0.0)	2 (2.0)	0.135
Cancer	15 (5.4)	11 (6.3)	4 (3.9)	0.583
Autoimmunological disease	13 (4.7)	4 (2.3)	9 (8.8)	0.018

**Table 2 diagnostics-12-00859-t002:** Haematological and biochemical variables.

Variables	Reference Values	Survivors		Non-Survivors		*p*-Value
		Mean ± SD	Med (iqr 25–75%)	Mean ± SD	Med (iqr 25–75%)	
RBC (10^6^/µL)	4.2–6.5	4.41 ± 0.56	4.40 (4.07–4.75)	4.31 ± 0.68	4.41 (3.83–4.80)	0.415
HB (g/dL)	12.0–18.0	13.04 ± 1.63	13.10 (12.10–14.10)	13.00 ± 2.05	13.40 (11.65–14.40)	0.648
HCT%	38.0–54.0	38.72 ± 4.93	39.00 (35.90–41.50)	38.67 ± 5.92	39.75 (34.60–43.00)	0.707
MCV fL	80.0–97.0	88.13 ± 6.24	89.00 (86.26–91.60)	90.08 ± 5.40	90.11 (86.75–93.73)	<0.01
MCH (pg/RBC)	26.0–32.0	29.61 ± 1.91	29.81 (28.69–30.90)	30.87 ± 6.56	30.27 (29.28–31.39)	<0.01
MCHC (g/dL)	31.0–36.0	33.45 ± 1.33	33.40 (32.51–34.30)	33.62 ± 1.39	33.55 (32.60–34.69)	0.346
RDW%	11.5–14.8	12.25 ± 1.41	12.10 (11.50–12.80)	12.65 ± 1.82	12.27 (11.41–13.43)	0.127
CRP (mg/L)	0.00–0.50	135.49 ± 97.81	121.60 (57.50–187.60)	158.98 ± 95.52	144.55 (87.25–231.30)	0.019
PCT (ng/mL)	0.17–0.35	0.89 ± 4.01	0.18 (0.10–0.40)	1.45 ± 3.69	0.30 (0.16–0.91)	<0.001
Creatine (mg/dL)	0.4–1.2	1.03 ± 0.84	0.84 (0.68–1.06)	1.25 ± 1.01	0.93 (0.69–1.35)	0.023
Vitamin D (ng/mL)	30–50	23.82 ± 12.95	21.8 (14.7–29.8)	22.37 ± 13.42	19.5 (13.4–30.5)	0.286
D-dimers (μg/L)	<500	6415 ± 20,486	1272 (762–2802)	17,559 ± 36,040	2180 (1189–13,951)	<0.001

Abbreviations: SD, standard deviation; med, median; iqr, interquartile range; RBC, red blood cells; HB, haemoglobin; HCT, haematocrit; MCV, mean cell volume; MCH, mean corpuscular haemoglobin; MCHC, mean corpuscular haemoglobin concentration; RDW, red cell distribution width; CRP, C-reactive protein; PCT, procalcitonin.

**Table 3 diagnostics-12-00859-t003:** White blood cell and platelet counts.

Variables	Reference Values	Survivors		Non-Survivors		*p*-Value
		Mean ± SD	Med (iqr 25–75%)	Mean ± SD	Med (iqr 25–75%)	
WBC (10^3^/µL)	4.0–10.2	10.26 ± 4.9	9.14 (7.13–12.60)	13.37 ± 6.93	11.98 (8.73–16.65)	<0.001
Neutrophils (10^3^/µL)	2.0–6.9	8.71 ± 4.59	7.39 (5.75–10.70)	11.62 ± 6.30	10.22 (7.50–14.72)	<0.001
Lymphocytes (10^3^/µL)	0.6–3.4	0.89 ± 0.43	0.78 (0.56–1.15)	1.70 ± 8.51	0.75 (0.53–1.03)	0.317
Monocytes (10^3^/µL)	0.00–0.90	0.56 ± 0.37	0.49 (0.33–0.68)	0.65 ± 0.49	0.54 (0.29–0.82)	0.405
Platelets (10^3^/µL)	140–420	285 ± 117	256 (205–343)	260 ± 120	242 (185–308)	0.071
NLR (10^3^/µL)	0.87–4.15	12.29 ± 9.43	9.33 (6.26–15.50)	17.70 ± 15.11	14.10 (9.10–20.22)	<0.001
PLR (10^3^/µL)	47–198	398 ± 256	318 (228–492)	395 ± 309	321 (204–462)	0.685
LMR (10^3^/µL)	2.45–8.77	1.97 ± 1.20	1.65 (1.13–2.49)	2.99 ± 8.28	1.55 (0.86–2.61)	0.211
SII (10^3^/µL)	142–808	3666 ± 3381	2507 (1435–4884)	4554 ± 4373	3512 (1994–5559)	0.031

Abbreviations: SD, standard deviation; med, median; iqr, interquartile range; WBC, white blood cells; NLR, neutrophil–lymphocyte ratio; PLR, platelet–lymphocyte ratio; LMR, lymphocyte–monocyte ratio; SII, systemic immune inflammation index.

**Table 4 diagnostics-12-00859-t004:** The relationships of the NRL and SII with the age and other inflammation markers.

	Variables	Age (Years)	CRP (mg/L)	PCT (ng/mL)
Survivors	NLR (10^3^/µL)	r_s_ = 0.218 *p* = 0.004	r_s_ = 0.191 *p* = 0.011	r_s_ = 0.249 *p* < 0.001
	SII (10^3^/µL)	r_s_ = 0.190 *p* = 0.011	r_s_ = 0.158 *p* = 0.035	r_s_ = 0.175 *p* = 0.019
Non-survivors	NLR (10^3^/µL)	r_s_ = 0.051 *p* = 0.597	r_s_ = 0.057 *p* = 0.558	r_s_ = 0.140 *p* = 0.149
	SII (10^3^/µL)	r_s_ = −0.039 *p* = 0.688	r_s_ = 0.011 *p* = 0.912	r_s_ = 0.096 *p* = 0.321

Abbreviations: r_s_, Spearman rank correlation coefficient.

**Table 5 diagnostics-12-00859-t005:** The statistical characteristics of the ROC curve for the univariate logistic model.

Variables	AUC	Cut-off	Sensitivity (%)	Specificity (%)	Predictive Value	
					Positive	Negative
Age	0.669	60.5	71.3	55.4	49.4	76.0
CRP	0.583	140.20	53.7	59.9	45.0	67.9
PCT	0.634	0.188	70.4	50.8	46.6	73.8
NLR	0.629	11.57	63.0	60.5	49.3	72.8
SII	0.576	2058	73.1	45.2	44.9	73.4

Abbreviations: AUC, the area under the curve; Cut-off, the optimal threshold value for clinical stratification.

**Table 6 diagnostics-12-00859-t006:** Univariate and multivariate Cox model analysis.

Variables	Univariate			Multivariate		
	HR	95% CI	*p*-Value	HR	95% CI	*p*-Value
Age	1.044	1.022–1.067	<0.001	1.041	1.018–1.063	<0.001
CRP	1.000	1.000–1.004	0.085			
PCT	1.296	1.116–1.505	0.001	1.212	1.043–1.408	0.012
NLR	1.317	1.004–1.728	0.047	2.122	1.219–3.694	0.008
SII	1.078	0.865–1.344	0.503	0.606	0.390–0.943	0.026

Abbreviations: HR, hazard ratio; 95% CI, confidence interval for the true population value of the HR.

## Data Availability

The data used to support the findings of this study are available from the corresponding author upon request.

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
