# Peer review of "Systemic Inflammatory Predictors of In-Hospital Mortality in COVID-19 Patients: A Retrospective Study"

_diagnostics, 2022, doi:10.3390/diagnostics12040859_

Round 1

Reviewer 1 Report

Minor comments

  1. There are numerous minor typographical errors throughout that need to be corrected.
  2. Please add p-values in Table 1 when comparing characteristics of survivor and non-survivor groups.
  3. Again, in Table 1, parameters should be capitalised throughout
  4. In Table 1, the following parameters are missing and should be added (or their exclusion explained/justified); BMI, current cigarette smoker, and N and % obese individuals in comorbidity section
  5. In Table 2, creatinine, serum vitamin D, D-Dimer and PaO2/FiO2 levels should be added (these variables have been shown in other studies to predict mortality in patients hospitalised for COVID-19 and need to be considered).

Major comments

My main criticism of this paper lies in the analysis performed. It appears that some key parameters are absent from the analysis that have previously been shown (from other studies) to predict mortality. Furthermore, a number of other studies have already explored the use of key clinical and laboratory measures to rapidly identify patients at risk of death. Potentially confounding variables also do not appear to have been considered in the analysis.

Fumagalli et al. 2020 developed a risk score based on clinical and laboratory parameters to identify  patients at risk of mortality and among a range of parameters evaluated, PaO2/FiO2, creatine, platelet count and respiratory rate emerged as key predictors.

On this basis, some of these respiratory parameters (oxygen saturation, respiratory rate and PaO2/FiO2) should be considered/addressed in the analysis since these too are also readily available and have previously been shown to identify these patients. Authors should present as many of these (“predictor”) variables in Table 1 and incorporate them in their subsequent analysis.

Furthermore, the role of vitamin D has been increasingly documented in several studies in patients with low serum (25-OH-D) levels (having greatest risk) compared to those with sufficient levels as well as benefit afforded in those receiving high doses of vitamin D (see Giannini et al. 2021). Also, in this paper, the number of comorbidities, high D-Dimer levels, current cigarette smokers (as well as vitamin D treatment) emerged as predictors of COVID-19-related death.  This is another example of additional variables that are worth incorporating in their analysis. The presentation of comorbidities would also be important to stratify as well by number of comorbidities; i.e. none, 1, 2, 3 or more to determine the role of this burden (as Giannini et al. described).

Therefore, I feel that in order to provide a more comprehensive screening of available parameters, the aforementioned should be considered in their analysis to determine their relative contribution.

In their analysis, authors identified predictor variables, namely CRP, PCT, NLR and SII (inverse) in multivariate analysis as being associated with mortality. Since (advanced) age remains a significant predictor of mortality, this variable (and several others as mentioned previously and also perhaps gender, smoking, creatine, comorbidities etc.) the tested variables (CRP, PCT, NLR, SII) need to be adjusted for these other (confounding variables) in order to understand how and to what extent they contribute to the risk of mortality independently. This analysis is currently missing. ROC analysis is useful but only subsequent to confirming that the 3 or 4 variables identified can independently predict mortality.    

Failing a robust “adjustment” of potential confounders, I believe that the analysis is not sufficiently robust in its current form. A limitations section should be prepared/expanded to address some specific variables that were not considered and/or justification for their lack of inclusion in the analysis.

Fumagalli C, et al. Clinical risk score to predict in-hospital mortality in COVID-19 patients: a retrospective cohort study. BMJ Open. 2020 Sep 25;10(9):e040729. doi: 10.1136/bmjopen-2020-040729. PMID: 32978207

Giannini S, et al. Effectiveness of In-Hospital Cholecalciferol Use on Clinical Outcomes in Comorbid COVID-19 Patients: A Hypothesis-Generating Study. Nutrients. 2021 Jan 14;13(1):219. doi: 10.3390/nu13010219. PMID: 33466642; PMCID: PMC7828675.

Author Response

Review 1

We greatly appreciate your time and effort dedicated to providing feedback on our manuscript and we are grateful for the insightful comments on and valuable improvements to our paper. All the suggestions helped us to evaluate our outcomes even more precisely in order to deliver improved, high quality scientific manuscript which we hope will now meet the high standards of Diagnostics.

Minor comments

  1. There are numerous minor typographical errors throughout that need to be corrected.

The typographical errors have been corrected in the whole manuscript.

  1. Please add p-values in Table 1 when comparing characteristics of survivor and non-survivor groups.

Table 1 has been supplemented with the suggested element

  1. Again, in Table 1, parameters should be capitalised throughout.

 Following the Reviewer’s suggestion, all parameters have been capitalised.

  1. In Table 1, the following parameters are missing and should be added (or their exclusion explained/justified); BMI, current cigarette smoker, and N and % obese individuals in comorbidity section.

The information concerning current cigarette smokers and obesity was collected on the basis of an interview with the family; this has been added in Table 1. The assessment of BMI was impossible due to a serious condition of patients on the day of admission to Intensive Care Unit; 38% patients died.

  1. In Table 2, creatinine, serum vitamin D, D-Dimer and PaO2/FiO2 levels should be added (these variables have been shown in other studies to predict mortality in patients hospitalised for COVID-19 and need to be considered).

The ratio of PaO2/FiO2 has been described in 2.1. Study population. The levels of creatinine, vitamin D, D-dimer have been included in Table 2 and described in 3.2.Haematological and biochemical variables.

Major comments

My main criticism of this paper lies in the analysis performed. It appears that some key parameters are absent from the analysis that have previously been shown (from other studies) to predict mortality. Furthermore, a number of other studies have already explored the use of key clinical and laboratory measures to rapidly identify patients at risk of death. Potentially confounding variables also do not appear to have been considered in the analysis. Fumagalli et al. 2020 developed a risk score based on clinical and laboratory parameters to identify patients at risk of mortality and among a range of parameters evaluated, PaO2/FiO2, creatine, platelet count and respiratory rate emerged as key predictors. On this basis, some of these respiratory parameters (oxygen saturation, respiratory rate and PaO2/FiO2) should be considered/addressed in the analysis since these too are also readily available and have previously been shown to identify these patients. Authors should present as many of these (“predictor”) variables in Table 1 and incorporate them in their subsequent analysis.

Thank you very much for this comment. The key clinical and laboratory measures to rapidly identify patients at risk of death from COVID-19 were analysed in a few papers [References 22-26]. We investigated platelets and platelet-to-lymphocyte ratio (PLR) but the differences were found to be insignificant between survivors and deceased patients (Table 3). The PaO2/FiO2 ratio was 121.45 ± 82.15 mm Hg for survivors and 104.28 ± 70.33 mm Hg for non-survivors (p>0.05), and the values have been described in 2.1. Study population.

Furthermore, the role of vitamin D has been increasingly documented in several studies in patients with low serum (25-OH-D) levels (having greatest risk) compared to those with sufficient levels as well as benefit afforded in those receiving high doses of vitamin D (see Giannini et al. 2021). Also, in this paper, the number of comorbidities, high D-Dimer levels, current cigarette smokers (as well as vitamin D treatment) emerged as predictors of COVID-19-related death. This is another example of additional variables that are worth incorporating in their analysis. The presentation of comorbidities would also be important to stratify as well by number of comorbidities; i.e. none, 1, 2, 3 or more to determine the role of this burden (as Giannini et al. described). Therefore, I feel that in order to provide a more comprehensive screening of available parameters, the aforementioned should be considered in their analysis to determine their relative contribution.

In the present study, we concentrated on systemic inflammatory markers in critically ill COVID-19 patients requiring mechanical ventilation in deep sedation. Most patients were dehydrated and required intensive fluid therapy in the initial phase of treatment. The other biochemical markers such as creatine, vitamin D and D-dimers have been measured and presented in Table 2. The high levels and a large statistical dispersion of D-dimer values were possibly related to long pre-hospital illness, dehydration, and noxious effects of SARS-CoV-2 on the endothelium.

Vitamin D was not administered to the patients; sustaining their lives was a priority. However, the study is currently being continued and the analyses have included a screening of inflammatory variables and immunomodulatory components of the diet in vaccinated and non-vaccinated COVID-19 patients.

In their analysis, authors identified predictor variables, namely CRP, PCT, NLR and SII (inverse) in multivariate analysis as being associated with mortality. Since (advanced) age remains a significant predictor of mortality, this variable (and several others as mentioned previously and also perhaps gender, smoking, creatine, comorbidities etc.) the tested variables (CRP, PCT, NLR, SII) need to be adjusted for these other (confounding variables) in order to understand how and to what extent they contribute to the risk of mortality independently. This analysis is currently missing. ROC analysis is useful but only subsequent to confirming that the 3 or 4 variables identified can independently predict mortality.    

The other biochemical variables were included into the basic statistical analysis (Table 2). However, Kaplan-Meier analysis, as a one-dimensional analysis, included age and systemic inflammatory variables, for which the ROC analysis was performed, and statistically significantly differentiated the course of the survival curves. All five predictors were included in the univariate analysis of the Cox model. Furthermore, multivariate analysis with the Cox model showed that the CRP predictor had no additional input as a predictor of patient survival (Akaike Information Criterion) when other factors (age, PCT, NLR, SII) were already included. This is partly the consequence of the fact that CRP is highly correlated with PCT (rs=0.585, p<0.001).

Failing a robust “adjustment” of potential confounders, I believe that the analysis is not sufficiently robust in its current form. A limitations section should be prepared/expanded to address some specific variables that were not considered and/or justification for their lack of inclusion in the analysis.

The Limitation section has been expanded according to the Reviewer’s suggestion.

Reviewer 2 Report

I have read the manuscript in a detailed fashion.  In this paper, the authors have tried to investigate whether routine blood tests and clinical characteristics can predict in-hospital mortality in COVID-19.  They showed that some simple parameters derived from complete blood cell analysis have prognostic implications in the course of COVID-19, enable the identification of patients at a higher risk of in-hospital mortality, and confirmate the preliminary observations on the suitability of neutrophils-to-lymphocytes ratio for prognostic stratification.

I think it is a good research paper and deserves to be published in this journal. However, the following minor comments concerns should be considered before publication:

*As is known, Kaplan-Meier method is a clever method of statistical treatment of survival times which not only makes proper allowances for those observations that are censored, but also makes use of the information from these subjects up to the time when they are censored.  However, a limitation of the KM method is that the log-rank test is purely a significance test and cannot provide an estimate of the size of the difference between the groups and its related confidence interval. Another limitation of the KM method is that it only provides unadjusted mortality (and survival) probabilities. Did the authors take this into account? The authors should make some comments or give some references about it, at least.  

* More comments may be added for Figures.

Author Response

Review 2

We greatly appreciate your time and effort dedicated to providing feedback on our manuscript and we are grateful for the insightful comments on and valuable improvements to our paper. All the suggestions helped us to evaluate our outcomes even more precisely in order to deliver improved, high quality scientific manuscript which we hope will now meet the high standards of Diagnostics.

Comments

I think it is a good research paper and deserves to be published in this journal. However, the following minor comments concerns should be considered before publication:

*As is known, Kaplan-Meier method is a clever method of statistical treatment of survival times which not only makes proper allowances for those observations that are censored, but also makes use of the information from these subjects up to the time when they are censored.  However, a limitation of the KM method is that the log-rank test is purely a significance test and cannot provide an estimate of the size of the difference between the groups and its related confidence interval. Another limitation of the KM method is that it only provides unadjusted mortality (and survival) probabilities. Did the authors take this into account? The authors should make some comments or give some references about it, at least.

Thank you very much for this comment. In order to present the risk of mortality we used KM method only for variables which significantly differed between survived and deceased patients. i.e. CRP and PCT (Table 2), NLR and SII (Table 3) Among the analysed variables, NLR demonstrated an impact on the mortality rate HR >2 (p<0.01). However, we were surprised by the results of CRP analysis. KM survival curves and Cox proportional hazard model analysis showed that CRP could not be used as an independent factor to predict the severity of COVID-19.

* More comments may be added for Figures.

Following the Reviewer’s suggestion, more comments have been added to the legend of figures.

Round 2

Reviewer 1 Report

Authors have addressed most of my comments but I would still like to see more discussion on findings from the two papers compfred to their study: 

Giannini et al. and Fumagalli et al.:

Fumagalli C, et al. Clinical risk score to predict in-hospital mortality in COVID-19 patients: a retrospective cohort study. BMJ Open. 2020 Sep 25;10(9):e040729. doi: 10.1136/bmjopen-2020-040729. PMID: 32978207

Giannini S, et al. Effectiveness of In-Hospital Cholecalciferol Use on Clinical Outcomes in Comorbid COVID-19 Patients: A Hypothesis-Generating Study. Nutrients. 2021 Jan 14;13(1):219. doi: 10.3390/nu13010219. PMID: 33466642; PMCID: 

1) The explanations provided for not incorporating parameters that were previously found to predict mortality in patients hospitilaised for COVID-19 are acceptable. However, I think that the Discussion needs to comment on the other scores and/parameters used to predict mortality from Fumagalli et al. and Giannini et al. papers (as already mentioned in the first review) in the context of the parameters used in the present study. A paragraph describing other scores and how they differ between the present study is required (i.e. the authors should compare patient characteristics between the studies to explain differences; age, gender, etc.).

D-Dimer levels wer 3-fold higher in non-survivors compared to survivors. Did the authors look at D-Dimer levels as a predictor of mortality? If not, please justify and also comment on D-Dimer levels from Giannini et al in the Discussion and also that while the majority of patients had levels that were borderline deficient (<20 ng/ml), vitamin D did not emerge as being different (between groups) in this study - but have been shown in several others etc..

2) I find the % patients with obesity extremely high and I would like the authors to comment on this (just under 50% were obese) and a slightly higher frequency of obesity in survivors vs. non-survivors (50 vs 43.8%). It may well be that a higher proportion of individuals with obesity are hospitalised - but the authors should comment on this - and that obesity, per se did not emerge as a predictor of mortality. Perhaps the fact that BMI was not used to identify patients as obese may have led to an overestimation?  How was obesity diagnosed? Is it possibile that the authors intended "overweight"?   

3) In the limitations section "long-rank" test should be corrected to "log-rank" test.  

Author Response

Review 1

We greatly appreciate the time spent on our manuscript revision. All of the comments motivated us to re-evaluate our outcomes in order to deliver an improved manuscript.

Major comments

The explanations provided for not incorporating parameters that were previously found to predict mortality in patients hospitilaised for COVID-19 are acceptable. However, I think that the Discussion needs to comment on the other scores and/parameters used to predict mortality from Fumagalli et al. and Giannini et al. papers (as already mentioned in the first review) in the context of the parameters used in the present study. A paragraph describing other scores and how they differ between the present study is required (i.e. the authors should compare patient characteristics between the studies to explain differences; age, gender, etc.).

Our study has reported the results obtained from 285 critically ill patients treated with invasive mechanical ventilation in the prone position at the intensive care unit (survivors n=177, non-survivors 108). All the cases were emergency admissions and partial pressure arterial oxygen to fraction of inspired oxygen (PaO2/FiO2) values were used as the major inclusion criterium. The patients were classified as severe acute respiratory distress syndrome (ARDS). The PaO2/FiO2 ratio was recorded at 121.45 ± 82.15 mm Hg in survivors and 104.28 ± 70.33 mm Hg in non-survivors (p>0.05). The excerpt in italics has been included in Materials and Method. The patients were relatively young: mean age of 62.0, with concomitant diseases including mainly obesity (47,7), hypertension (55,2) and diabetes (20,7). By contrast, the study by Fumagalli et al. the value of PaO2/FiO2 below 200 was observed in 101(19,6) patients (out of the total of 516). The mean age of that study population was 67 yrs, 79 yrs in the deceased patients and 64 yrs in the survivors (p-value <0,001). In the patients managed with invasive ventilation 59(15,0) patients survived and 42(35,0) patients died, p-value <0,001. The conditions which prevailed in the concomitant diseases included smoking history (21,7), hypertension (35,3), cardiovascular diseases (28,5) and depression (20,1). On balance:  although our study patients were younger than the ones in the study by Fumagalli et al., their condition on admission was markedly more severe and all of them required mechanical ventilation.  This is likely to be the consequence of a considerable delay in a call for medical help or a limited access to such service, however, such data were mostly impossible to verify.  

D-Dimer levels were 3-fold higher in non-survivors compared to survivors. Did the authors look at D-Dimer levels as a predictor of mortality? If not, please justify and also comment on D-Dimer levels from Giannini et al in the Discussion and also that while the majority of patients had levels that were borderline deficient (<20 ng/ml), vitamin D did not emerge as being different (between groups) in this study - but have been shown in several others etc.

High levels of D-Dimer observed in non-survivors (mean (mg/L ) survivors 6415 versus non-survivors 17559) were consistent  with the outcomes reported in a large number of previous studies including the paper by Giannini et al. In our study, we considered the significance of D-Dimer level as an acknowledged predictor of mortality and we focused on the inflammatory calculators as the major subject of our analysis.

I find the % patients with obesity extremely high and I would like the authors to comment on this (just under 50% were obese) and a slightly higher frequency of obesity in survivors vs. non-survivors (50 vs 43.8%). It may well be that a higher proportion of individuals with obesity are hospitalised - but the authors should comment on this - and that obesity, per se did not emerge as a predictor of mortality. Perhaps the fact that BMI was not used to identify patients as obese may have led to an overestimation?  How was obesity diagnosed? Is it possible that the authors intended "overweight"? 

BMI evaluation on admission was hampered by a serious general condition of the patient as it was impossible to make precise measurement of their weight. The degree of obesity was assessed by medical personnel on the basis of the patient history taking, visual assessment and then complemented with the information obtained from the patient’s family. That is why our initial version of the paper did not consider these data as they were likely not to be objective enough and therefore, we did not use obesity as a predictor of mortality in the study population. No significant changes were observed in Vitamin D level between the survivors and the deceased patients. Such an observation may potentially be linked to the patients’ age or chronic Vitamin deficiency in Polish population. This aspect was not investigated and analysed in our paper.  
